# Phytotoxicity of Extracts of *Argemone mexicana* and *Crotalaria longirostrata* on Tomato Seedling Physiology

**DOI:** 10.3390/plants12223856

**Published:** 2023-11-15

**Authors:** Henry López López, Mariana Beltrán Beache, Yisa María Ochoa Fuentes, Ernesto Cerna Chavez, Epifanio Castro del Ángel, Juan Carlos Delgado Ortiz

**Affiliations:** 1Parasitología Agrícola, Universidad Autónoma Agraria Antonio Narro, Calzada Antonio Narro, Saltillo C.P. 25315, Mexico; henry.lopez.l@hotmail.com (H.L.L.); jabaly1@yahoo.com (E.C.C.); epifaniocastrodelangel@hotmail.com (E.C.d.Á.); 2Centro de Ciencias Agropecuarias, Epartamento de Fitotecnia, Universidad Autónoma de Aguascalientes, Aguascalientes C.P. 20700, Mexico; 3Consejo Nacional de Humanidades, Ciencia y Tecnología, Ciudad de México 03940, Mexico

**Keywords:** agronomic variables, arvense, alkaloids, bioactive compounds, ethnobotany

## Abstract

Phytotoxicity caused by secondary metabolites of botanical extracts is a drawback in agriculture. The objective of this study was to evaluate the phytotoxic effects of methanolic extracts of *Crotalaria longirostrata* and *Argemone mexicana* on the germination and physiological variables of tomato seedlings. The results indicated that high doses of both extracts (*Clong500* and *Amex500*) inhibited tomato seed germination, while their mixture (*Cl50* + *Am50*) promoted germination by 100%. At 30 days after transplanting (dat), the plant height increased by 15.4% with a high dose of *C. longirostrata* (*Clong500*) compared to the control. At 30 dat, the vigor index displayed a notable increase with *Cl50* + *Am50*, reaching 29.5%. The root length increased with the mean dose of *A. mexicana* (*Amex95*) at 10, 20, and 30 dat (59.7%, 15.1%, and 22.4%, respectively). The chlorophyll content increased with *Amex95* by 66.1% in 10 dat, 22.6% at 20 dat, and 19.6% at 30 dat. On the other hand, *Amex95* had a higher nitrogen content throughout the trial. *Amex95* produced the greatest increase in root dry weight by 731.5% and 209.4% at 10 and 20 dat. The foliage dry weight increased by 85.7% at 10 dat with *Amex95* and up to 209.7% with *Amex50* at 30 dat. The present investigation reveals the ability of the extracts to stimulate tomato growth at low and medium doses, though at high doses they exhibit allelopathic effects.

## 1. Introduction

Conventional agriculture focuses on the use of chemical pesticides (an estimated 2.5 million tons of synthetic active ingredients are used annually) to rapidly control pests and diseases of major crops produced worldwide. However, these pesticides reduce the diversity of non-target species, such as natural enemies of pests, and have toxic effects on ecosystems and human health [1,2]. This is due to direct or indirect exposure to synthetic active ingredients that generate negative ecological impacts, in addition to risks to agricultural production due to damage to the soil microbiome [3].

To counteract the harmful effects generated by conventional agriculture, innovations in pest and disease management have emerged through agroecological systems for sustainable production that promote non-curative measures through preventive methods employing mechanical control and the use of biological or botanical pest control agents; these systems will limit the use of chemical pesticides through a spatial and temporal organization of agronomic and ecological concepts that avoid socioecological vulnerability [4,5].

For this reason, using plant extracts as an alternative to chemical pesticides is one solution, as these extracts have proved to be suitable alternatives for weed control, plant biostimulants, pest control, and the management of agricultural diseases [6,7].

Every year about 2000 plant species with a potential for pest and disease control in crops are discovered (as of 2015, 400,000 plant species were registered), and the use of plants as pesticides is possible due to the presence of secondary metabolites. There are more than 200,000 reported compounds among non-protein amino acids, anthocyanins, alkaloids, amines, glucosinolates, cyanogenic glycosides, terpenoids, phenylpropanoids, flavonoids, tannins, polyketides, saponins, coumarins, organic acids, and carbohydrates [8,9]. It is believed that plant secondary metabolites do not induce resistance, cause low harmful effects to the environment, are safer for human health, and are more economical for farmers [10].

One plant that can be used for this purpose is *Crotalaria longirostrata* Hook. & Arn. (Fabaceae), colloquially known as chipilín, which is used as a food in southeastern regions of Mexico and Central America [11]. The incorporation of *C. juncea* species into the substrate as solid fertilizer for crops such as beans and maize can increase nitrogen mineralization by up to 85% and carbon availability by 80% [12]. Similarly, *C. retusa* leaf powder, used to grow *Solanum lycopersicum*, increased leaf biomass by 24.7% and root biomass by 34.1% [13].

However, allelopathic effects have also been reported for some species of this genus, such as *C. retusa* (used in a methanolic extract), which exhibits phytotoxic mechanisms at high doses of crude alkaloids on *Phaseolus vulgaris*, reducing germination by 69.3%, total protein by 81.7%, and leaf sugar by 62.3% [14].

Another plant with similar characteristics is the species *Argemone mexicana* Linn (Papaveraceae), known as chicalote, an annual weed present in Mexico and used as an indigenous medicinal plant [15,16]. *A. mexicana* has demonstrated that it contains adequate levels of nitrogen and potassium for use as an organic fertilizer once it has been converted to vermicompost [17]. On the other hand, a phytotoxic effect was reported when *A. mexicana* was used in a 1% *w/v* aqueous extract, inhibiting tomato and lettuce root germination and growth, while a 1% *w/v* methanolic extract reduced tomato root elongation by 48% [18].

Secondary metabolites present in plant extracts proved to be fundamental components in the induction of plant resistance to pathogen attack [19]. However, their capacity to induce plant growth and improve the yield of certain crops is not well known [20]. One example is saturated fatty acids, such as unsaturated fatty acids found in extracts of various organs of *Moringa oleifera*, which promote seed germination and plant development of *Triticum aestivum* L. [21]. Aqueous extracts of *Posidonia oceanica* revealed that phenolic compounds, glycosides, and terpenes played a crucial role in the development of the roots and leaves of *S. lycopersicum* and *Cucumis sativus* [22]. The methanolic extracts of *C. longirostrata* and *A. mexicana* used in this study contain secondary metabolites previously identified as significant components in these plants [23,24].

Although there are several studies on the effect of the *A. mexicana* species [25,26] and the genus *Crotalaria* [12,13,27] as plant growth stimulants, evidence of the effects of such plants is scarce. Therefore, the present work aimed to evaluate the phytotoxic effects of methanolic extracts of *Crotalaria longirostrata* and *Argemone mexicana* on the germination and physiological variables of tomato seedlings.

## 2. Results

### 2.1. Effect of Extracts on Tomato Seed Germination

The germination of tomato seeds after the application of methanolic extracts of *C. longirostrata* and *A. mexicana* (Table 1) was sensitive to high doses of both extracts (*Clong500* and *Amex500*); the highest germination was observed with the mixture of the extracts *Cl50* + *Am50*, with 100% germination; a promotion of germination was noted in comparison with the germination rate observed in the control seeds.

### 2.2. Greenhouse Test

At the beginning of the study, the tomato plants treated with the methanolic extracts were smaller than the control plants (Table 2); conversely, development responded favorably at 20 dat, showing greater growth with *Clong500* and *Amex95*. Better results were obtained at 30 dat with the mixture of extracts (*Cl50* + *Am50*), with growth increases of 14.9% and 15.4% with *Clong500*.

After applying methanolic extracts to tomato plants at 10 dat, all treatments expressed a low vigor index (Table 3). At 20 and 30 dat, a slight increase in vigor index was recorded in the *Amex95* treatment (4.5% and 6.8%) and a moderate increase in the *Clong50* treatment (8.4% and 8.9%). After 30 dat, the *Amex50* dose reached a vigor index of 10.7%, while the mixture of extracts (*Cl50* + *Am50*) achieved the highest increase with 29.5%.

As shown in Table 4, treatment with *Amex95* stimulated an increased root growth of the tomato plants throughout the trial (10.7, 17.5, and 29.5 mm, respectively). However, at 20 days post transplanting, the *Clong500* treatment showed a similar increase (17.6 mm) to the *Amex95* treatment. At the end of the trial, the *Amex95* and *Amex500* treatments similarly increased root length by 29.5 mm and 29.3 mm.

The chlorophyll content in tomato leaves was higher with methanolic extracts (Figure 1A), and the chlorophyll content values for the first 10 dat were higher in *Clong500* and *Amex95* (44.9 and 45.1 SPAD, respectively). At 20 dat, the SPAD values increased in all treatments compared to the control, but the value was higher in all treatments compared to the first evaluation; the best treatments were *Clong500* with 58.5 SPAD and *Amex95* with 58.7 SPAD. At 30 dat, the increase was 67.7 and 67.6 SPAD with *Clong95* and *Amex50*. Like chlorophyll, the nitrogen content increased at 10 dat in the *Clong500* (14.4) and *Amex95* (14.1) treatments; after 20 dat, it increased by 18.6 and 18.7, respectively. At 30 dat, the *Clong95* (21.6) and *Amex50* (21.4) treatments showed the highest nitrogen levels (Figure 1B).

After the application of *Amex95* at 10 dat, the plant root dry weight increased by 0.74 g compared to the control (Figure 2A). Subsequently, at 20 dat the *Amex95* treatment increased the root dry weight by 1.64 g, and *Clong50* increased it by 1.69 g. At 30 days after transplanting, the *Amex50* and *Amex500* treatments showed a very similar increase in root dry weight (2.6 and 2.62 g, respectively), while *Clong95* was superior with 2.75 g. Regarding the dry weight of the aerial part of the plant after receiving different doses of the extracts, the dry biomass showed a significant increase (Figure 2B). Through treatment with *Amex95* it reached 0.65 g after 10 dat, and *Amex500* obtained a dry biomass of 4.1 g at 20 dat. At the end of the trial (30 dat), *Amex50* showed the greatest increase in dry biomass with 21.65 g, followed by *Amex95* (20.86 g) and *Amex500* (20.85 g).

## 3. Discussion

Plant extracts derived from roots, leaves, stems, flowers, and seeds are sources of secondary metabolites that may produce stimulant or allelopathic effects [7]. They are considered as an option to reduce losses in agriculture, given that when applied to leaves or seeds, they improve nutrient absorption capacities and strengthen resistance to abiotic and biotic stresses [6].

A metabolite with phytotoxic activity must become absorbed by the different plant tissues translocated through the phloem and xylem; therefore, its mode of action depends on the accepting plant species and the resources it limits (water, nutrients, light, ATP synthesis, gene expression, or damage to the cell cycle) [28].

Although studies on the possible phytotoxic properties of *C. longirostrata* are limited, other species of the *Crotalaria* genus have demonstrated allelopathic effects on seed germination. The aqueous extract of *C. juncea* leaves, at 1.5 and 3.3% *w*/*v*, significantly affected tomato seed germination by 12 and 100% [29]. Also, a methanolic extract of *C. juncea* leaves (1 mg/mL) inhibited the germination of *Vigna radiata* by 50% [30]. The extract containing chloroform fraction and leaves of *C. retusa* at concentrations of 10, 50, and 100 µg/mL reduced the germination ability of *P. vulgaris* by up to 60% [14], and a 15% (*w*/*v*) aqueous extract of *C. juncea* leaves inhibited the germination of *Zea mays* by up to 56.1% [31]. However, the methanolic extract of *C. juncea* roots promoted up to 40% more germination [30].

The aqueous extract at 1% (*w*/*v*) of *A. mexicana* reduced the germination rate of tomato seeds by 90% [18]. In Sorghum bicolor seeds, a 14 and 79.7% reduction in germination was observed when the aqueous extract was applied at doses of 5 and 25% (*w*/*v*) [32]. Furthermore, an aqueous extract of *A. mexicana* at 50 g/L inhibited the germination of *Brachiaria dictyoneura* by 47.4% and that of *Clitoria ternatea* by 20.5% [33]. The results obtained in this study showed that *Clong500* and *Amex500* inhibited tomato germination by up to 8.9%, and conversely, a mixture of the two extracts stimulated germination by up to 100% (Table 1).

Aqueous extracts of the leaves of *C. brevidens*, *C. sessiliflora*, and *C. juncea* (*w*/*v*) increased the growth of *Triticum aestivum* by 7.1 to 12.7% [34]. The aqueous leaf extract of *C. juncea* (*w*/*v*) increased the stem height of *Lactuca sativa* by 7.7% [35]. Likewise, a 50% *w*/*v* aqueous extract of *A. mexicana* stems stimulated the height of tomato plants by 18.5% [25]. Namkeleja et al. reported that the aqueous extract of *A. mexicana* at 50 g/L promoted the development of *B. dictyoneura* (55.3%) and *C. ternatea* (32.5%) plants [33]. These results are consistent with those observed in Table 2, where tomato plant height was greater when *C. longirostrata* and *A. mexicana* extracts were applied.

The seed vigor index indicates the relationship between environmental or genetic factors affecting seed quality, rapid and uniform plant emergence, plant weight, and/or plant growth rate under field or greenhouse conditions [36,37]. As can be seen in Table 3, the highest vigor index was presented with *Cl50* + *Am50*, suggesting that the extracts improve tomato seed and plant vigor.

There are several *Crotalaria* species (*C. brevidens*, *C. juncea*, *C. lanceolata*, *C. pallida*, *C. sessiflora*, and *C. spectabilis*) that have demonstrated their allelopathic effect when used as ground cover on *T. aestivum*, affecting root growth by up to 40.6% and the length of the longest root by up to 53.7% [34]. However, an aqueous extract of *C. juncea* leaves at a low dose (7.5% *w*/*v*) increased the length of the longest root of *Z. mays* by 9.2% [31]; the use of *C. juncea* as a ground cover (500 g/m^2^) increased the root development of *Z. mays* by 11.2% 24 days after emergence (dae) [38]. The present results showed a similar increase in root length after *Amex95* and *Amex500* treatments.

A report showed that chlorophyll content increased when *C. juncea* species were applied as green manure in the *Z. mays* crop, by up to 35.6% at 56 dde [39], and in *Oryza sativa* by up to 25.5% [40]. As shown in Figure 1, the increases in chlorophyll and nitrogen are closely related, and the treatments of *C. longirostrata* and *A. mexicana* also yielded a positive effect on tomato leaves.

Regarding the dry weight, the results of the methanolic extract of *A. mexicana* increased the dry weight of the root and aerial part of the tomato plant compared to the extract of *C. longirostrata* (Figure 2). In *L. sativa* plants, there was an increase of 16.6% due to the use of the aqueous extract of *C. juncea* roots at 50 g/L [35], and an increase of up to 112.3% was observed in *O. sativa* plants at 60 days after emergency (dae) when using *C. juncea* as a ground cover [40]. Likewise, a 50% (*w*/*v*) aqueous extract of *A. mexicana* stems increased the total dry weight of tomato plants by 38% [25].

In addition, the results of gas chromatography and mass spectrophotometry showed the presence of secondary metabolites in the extracts, which could be responsible for their capacity to stimulate plant growth. The methanolic extracts of *A. mexicana* and *C. longirostrata* presented a 72.6% and 79.9% abundance of these compounds, respectively; these metabolites belong to the groups of saturated and unsaturated fatty acids, amines, fatty alcohols, alkaloids, and cyanogenic glycosides [23,24].

Metabolites such as saturated and unsaturated fatty acids may promote plant growth for the synthesis of cellular constituents [41,42]. These lipids act in the tricarboxylic acid cycle, contributing to enzyme activation, and may increase germination, root formation, photosynthesis, trichome number, and fruit yield [43,44]. In the case of fatty acids, they are involved in jasmonic acid biosynthesis, fruit ripening, tuber formation, and pollen development [45,46].

Alkaloids are a group of secondary metabolites with a varied structure, mainly consisting of two carbon and nitrogen rings, with substituent groups at carbon 1 and 7 [8]. The various modifications that nitrogen presents in the molecule (heterocyclic and non-heterocyclic) allow for arrangements in the chemical structure and its biological activity [47]. Due to their carbon and nitrogen composition, alkaloids could potentially contribute to plant nutrition [48].

It is essential to consider that the secondary metabolites present in plant extracts can induce toxicity in the plant. The effects of low doses of the stimulus utilized can trigger an adaptive response in the organism, while high doses cause an increase in the organism’s resistance or inhibition, referred to as hormesis [49]. Hormesis establishes a dose–time–response connection that can vary depending on the individual and its characteristics, generating protective responses in plants by stimulating cellular defense mechanisms due to low doses or stress caused by high doses [50]. The application of hormesis in plant growth and resistance using plant extracts faces significant challenges due to the need for more comprehensive investigations on the inheritance of hormesis transgenerational effects (HTE) across multiple plant generations [51]. These studies should address the underlying epigenetic mechanisms that may generate phenotypic variability and the generation of new heritable epialleles in subsequent generations [52].

In addition, it has been observed that plants experience a hormetic response to low doses of exogenous agents for short periods, which drives photosynthesis through photosystem II; this process is activated by reducing the quenching of non-photochemical fluorescence, which dissipates excess energy and maintains a basal level of reactive oxygen responses [53,54]. This phenomenon accounts for the positive increase in chlorophyll levels in the tomato plants treated with the methanolic extracts applied in this study (Figure 1A). Furthermore, plant secondary metabolites act as allelochemicals, stimulating plant growth at low doses, and acting as inhibitors at high doses [55]. The effect of hormesis in plants, caused by applying plant extracts, was reported at a rate of 0.4%, in which 70% of the results obtained were favorable in plant growth, while 18% affected the increase in plant metabolism [56]. The aqueous extract of *M. oleifera* has generated the adaptability of 12 secondary metabolites at a low dose of 2.5% (*w*/*v*); these metabolites are responsible for promoting *Lepidium sativum* shoot growth by 48%, but, at high doses of 10% (*w*/*v*), a significant inhibition in root length of 85% and shoot length of 38% was observed [57].

These results indicate that, when considering using plant extracts as plant growth promoters, it is necessary to consider studies that assess ecological risks. Moreover, it is required to investigate selective hormesis in populations (including plants, fungi, and insects), to apply statistical models that demonstrate the adaptive response of hormesis (characterized by a biphasic dose–response), and to analyze the impact on international regulations [49,58].

## 4. Materials and Methods

The authors of this study evaluated the experiment in the laboratory and greenhouse of the Parasitology Department of the Universidad Autónoma Agraria Antonio Narro in Saltillo, Coahuila, Mexico.

### 4.1. Sampling of Plants and Preparation of Extracts

The species *C. longirostrata* (Hook. & Arn.) and *A. mexicana* (Linn) have been taxonomically identified and included in the ANSM herbarium of the Universidad Autónoma Agraria Antonio Narro with record numbers 104040 and 103807, respectively.

For *C. longirostrata*, plant samples were collected in Chiapa de Corzo, Chiapas, Mexico, and only the leaves were dried in the shade for seven days. They were then milled in a blender (model 7011s, Waring Commercial, Torrington, CT, USA), left to macerate for 30 days (0.2 g dry matter/mL of 96% methanol), and finally filtered on Whatman paper No. 1 (Sigma-Aldrich, St. Louis, MO, USA) [59]. *A. mexicana* was collected in Saltillo, Coahuila, Mexico, obtaining only leaves, dried in the shade for ten days, and pulverized in a blender (model 7011s, Waring Commercial, Torrington, CT, USA). Maceration was carried out for seven days with continuous agitation at room temperature (0.1 g dry matter/mL of 96% methanol) and filtered through Whatman paper No. 1 [60]. The two extracts were stored in dark flasks at 4 °C until use.

### 4.2. Plant Material

We used tomato (*Solanum lycopersicum*) seeds of the Rio Grande variety. The tomato seeds were immersed in a 6% sodium hypochlorite solution for 10 s, and then washed at least five times with sterile distilled water. We repeated this process twice to remove pesticide residues and disinfect the seeds.

### 4.3. Effect of Extracts on Tomato Seed Germination

The preparation of the concentrations followed that described by [23], where the relative density of each extract was obtained using a 25 mL Gay-Lussac pycnometer (16038, BRAND, Wertheim, Germany), and the density of water was taken as 0.997299 g/cm^3^ at 24 °C for *C. longirostrata* and 0.997772 g/cm^3^ at 22 °C for *A. mexicana*.

The lethal concentrations (LC) of *C. longirostrata* [23] and *A. mexicana* [24] have come from previous studies. Based on these concentrations, the treatments evaluated were for *C. longirostrata* an LC_50_ of 4.78 mg/mL (*Clong50*) and an LC_95_ of 14.52 mg/mL (*Clong95*), for *A. mexicana* an LC_50_ of 7.63 mg/mL (*Amex50*) and a CL_95_ of 107.98 mg/mL (*Amex95*), the doses of each extract at 500 mg/mL (*Clong500* and *Amex500*), and the mixture of extracts with the CL_50_ of each (*Cl50* + *Am50*). To evaluate the phytotoxic or growth-stimulant effects on tomato plants.

For testing seed viability, 15 seeds were placed in each Petri dish with filter paper, which was moistened with 10 mL of the different concentrations of the methanolic extracts of *C. longirostrata* and *A. mexicana*, while the control was distilled water [61]. Four replicates were established for each treatment and the absolute control. The Petri dishes were kept in the laboratory at a temperature of 25 °C, and the germinated seeds were counted after eight days (germinated seeds were considered those with visible radicle and hypocotyl). The test had a completely randomized design and four replications. The results were expressed in terms of germination percentage:$$\%\;germination=\frac{X}{Y}*100$$
where *X* is the number of seeds germinated in each treatment at eight days and *Y* is the number of seeds germinated in the absolute control at eight days.

### 4.4. Greenhouse Assay

Tomato seeds were germinated in 200-cavity polystyrene trays with peat moss as substrate and transplanted 30 days after emergence. Subsequently, they were placed in 10 L pots containing a mixture of perlite and peat moss (1:1). Plant nutrition was carried out with Steiner’s nutrient solution [62], according to the phenological stage of the crop. Different doses of methanolic extracts were applied by spraying with the aid of an atomizer. The first application was made immediately after transplanting, the second application was made 10 days after transplanting (dat), and the third was applied at 20 dat.

The vigor index is a quality parameter that considers the germination capacity and viability of the same batch of seeds, and is calculated according to the formula [61]:Vigor Index=G∗L
where *G* is the germination rate obtained in the laboratory test and *L* is the length of the aerial part of the plant obtained in the greenhouse.

The plant aerial height in cm was recorded with a flexometer, the root length in mm was measured with a Vernier caliper, and the plant dry weight (root and foliage) in g was obtained with an analytical balance (Ohaus, Parsippany, NJ, USA); the chlorophyll index and nitrogen content were measured using the Minolta SPAD 502 plus a chlorophyll meter (Konica Minolta Holdings Co., Ltd., Tokyo, Japan) (which provides chlorophyll in SPAD units and the value of nitrogen present in the leaf) [63]. Readings for all variables were taken at 10, 20, and 30 dat.

### 4.5. Statistical Analysis

The obtained data were processed with an analysis of variance, and means were compared with Tukey’s test (*p* ≤ 0.05) under a completely randomized design. All analyses were performed with the Statistical Analysis System (SAS) version 9.0 statistical software.

## 5. Conclusions

We found that methanolic extracts of *A. mexicana* and *C. longirostrata* leaves at high doses showed phytotoxic activity on *S. lycopersicum*, which affected germination, plant height, and vigor index. On the other hand, the two combined extracts showed the ability to stimulate seed germination. These methanolic extracts showed active properties as a plant growth promoter in *S. lycopersicum* cultivation and as a possible insecticide, suggesting their application to reduce the use of synthetic molecules.

## Figures and Tables

**Figure 1 plants-12-03856-f001:**
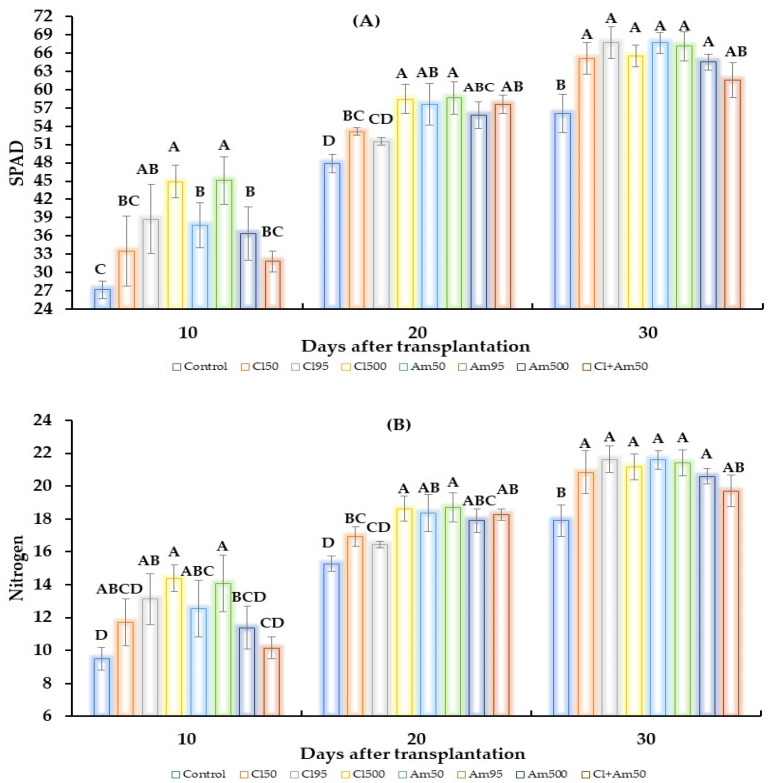
Effect of methanolic extracts of *C. longirostrata* and *A. mexicana* on tomato plants. (**A**) Chlorophyll content, and (**B**) nitrogen content in the leaves of the tomato plants treated with the extracts. The capital letters in each bar indicate statistical difference (ANOVA, Tukey HSD, *p* ≤ 0.05). Control refers to the absolute control group, *Clong50* stands for the CL_50_ of *C. longirostrata*, *Clong95* = represents the CL_95_ of *C. longirostrata*, *Clong500* corresponds to a concentration of *C. longirostrata* of 500 mg/mL, *Amex50* stands for the CL_50_ of *A. mexicana*, *Amex95* represents the CL_95_ of *A. mexicana*, *Amex500* corresponds to a concentration of *A. mexicana* of 500 mg/mL, and *Cl50 + Am50* refers to a mixture of the CL_50_ of both extracts.

**Figure 2 plants-12-03856-f002:**
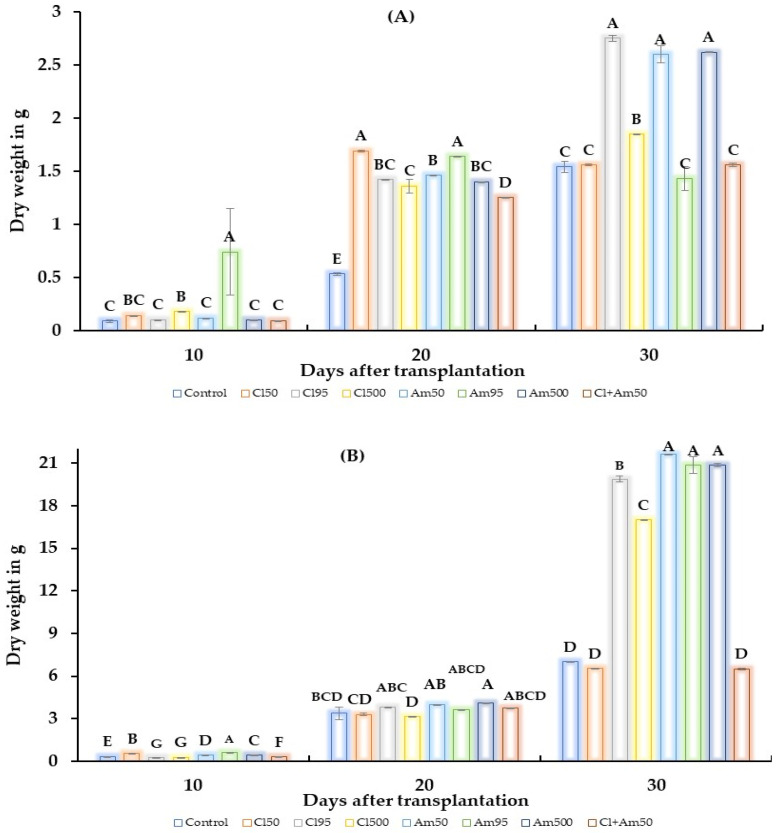
Effect of methanolic extracts of *C. longirostrata* and *A. mexicana* on tomato plants. (**A**) Leaf dry weight response, and (**B**) root dry weight response of tomato plants treated with the extracts. The capital letters in each bar indicate statistical difference (ANOVA, Tukey HSD, *p* ≤ 0.05). Control refers to the absolute control group, *Clong50* stands for the CL_50_ of *C. longirostrata*, *Clong95* represents the CL_95_ of *C. longirostrata*, *Clong500* corresponds to a concentration of *C. longirostrata* of 500 mg/mL, *Amex50* stands for the CL_50_ of *A. mexicana*, *Amex95* represents the CL_95_ of *A. mexicana*, *Amex500* corresponds to a concentration of *A. mexicana* of 500 mg/mL, and *Cl50 + Am50* refers to a mixture of the CL_50_ of both extracts.

**Table 1 plants-12-03856-t001:** Effect of methanolic extracts of *C. longirostrata* and *A. mexicana* on tomato seed germination.

Treatments	% Germination
Control	88.9 ± 3.81 bc
*Clong50*	91.1 ± 3.81 ab
*Clong95*	86.7 ± 0.15 bc
*Clong500*	80.0 ± 6.70 c
*Amex50*	86.7 ± 0.10 bc
*Amex95*	86.7 ± 0.15 bc
*Amex500*	84.47 ± 3.87 bc
*Cl50 + Am50*	100 ± 0 a
*p*-value	0.0001

According to Tukey’s test (*p* ≤ 0.05), means with the same letters are not significantly different. Control refers to the absolute control group, *Clong50* stands for the CL_50_ of *C. longirostrata*, *Clong95* represents the CL_95_ of *C. longirostrata*, *Clong500* corresponds to a concentration of *C. longirostrata* of 500 mg/mL, *Amex50* stands for the CL_50_ of *A. mexicana*, *Amex95* represents the CL_95_ of *A. mexicana*, *Amex500* corresponds to a concentration of *A. mexicana* of 500 mg/mL, and *Cl50 + Am50* refers to a mixture of the CL_50_ of both extracts.

**Table 2 plants-12-03856-t002:** Response of tomato plant height to methanolic extracts of *C. longirostrata* and *A. mexicana*.

Treatments	Height (cm) Days after Transplantation		
	10	20	30
Control	11.25 ± 1.75 a	18.83 ± 1.44 ab	22.30 ± 0.96 b
*Clong50*	10.08 ± 1.32 ab	19.80 ± 2.02 a	23.67 ± 1.26 ab
*Clong95*	8.42 ± 0.86 b	18.67 ± 0.58 ab	22.60 ± 0.36 ab
*Clong500*	9.70 ± 1.64 ab	20.00 ± 1.73 a	25.73 ± 2.16 a
*Amex50*	9.75 ± 1.69 ab	19.16 ± 1.62 ab	25.30 ± 0.87 ab
*Amex95*	11.08 ± 0.58 a	20.17 ± 1.44 a	24.40 ± 0.53 ab
*Amex500*	10.27 ± 0.98 ab	17.83 ± 0.29 ab	24.60 ± 0.36 ab
*Cl50 + Am50*	9.08 ± 0.66 ab	15.70 ± 0.17 b	25.63 ± 1.56 a
*p*-value	0.0066	0.0162	0.0122

Means with the same letter in the same column are not significantly different by Tukey’s test (*p* ≤ 0.05). Control refers to the absolute control group, *Clong50* stands for the CL_50_ of *C. longirostrata*, *Clong95* represents the CL_95_ of *C. longirostrata*, *Clong500* corresponds to a concentration of *C. longirostrata* of 500 mg/mL, *Amex50* stands for the CL_50_ of *A. mexicana*, *Amex95* represents the CL_95_ of *A. mexicana*, *Amex500* corresponds to a concentration of *A. mexicana* of 500 mg/mL, and *Cl50 + Am50* refers to a mixture of the CL_50_ of both extracts.

**Table 3 plants-12-03856-t003:** Vigor index of tomato plants treated with methanolic extracts of *C. longirostrata* and *A. mexicana*.

Treatments	Plant Vigor Index Days after Transplantation		
	10	20	30
Control	996.7 ± 114.7 a	1672.2 ± 108.6 a	1979.8 ± 13.1 b
*Clong50*	918.3 ± 74.6 abc	1812.2 ± 255.9 a	2156.7 ± 156.2 ab
*Clong95*	729.4 ± 62.6 c	1617.8 ± 50.1 a	1958.7 ± 31.2 b
*Clong500*	774.0 ± 87.7 bc	1606.7 ± 257.9 a	2068.6 ± 345.2 b
*Amex50*	845.0 ± 86.7 abc	1652.4 ± 140.1 a	2192.6 ± 75.1 ab
*Amex95*	960.6 ± 33.1 ab	1747.8 ± 125.1 a	2114.6 ± 45.9 b
*Amex500*	867.2 ± 69.1 abc	1505.6 ± 60.8 a	2076.4 ± 66.8 b
*Cl50 + Am50*	908.3 ± 38.2 abc	1570.0 ± 17.3 a	2563.3 ± 155.7 a
*p*-value	0.007	0.034	0.005

Means with the same letter in the same column are not significantly different by Tukey’s test (*p* ≤ 0.05). Control refers to the absolute control group, *Clong50* stands for the CL_50_ of *C. longirostrata*, *Clong95* represents the CL_95_ of *C. longirostrata*, *Clong500* corresponds to a concentration of *C. longirostrata* of 500 mg/mL, *Amex50* stands for the CL_50_ of *A. mexicana*, *Amex95* represents the CL_95_ of *A. mexicana*, *Amex500* corresponds to a concentration of *A. mexicana* of 500 mg/mL, and *Cl50 + Am50* refers to a mixture of the CL_50_ of both extracts.

**Table 4 plants-12-03856-t004:** Response of tomato plant root length to methanolic extracts of *C. longirostrata* and *A. mexicana*.

Treatments	Root Length (mm) Days after Transplantation		
	10	20	30
Control	6.7 ± 0.35 c	15.2 ± 1.63 b	24.1 ± 0.10 b
*Clong50*	9.2 ± 0.49 ab	15.5 ± 0.23 ab	26.5 ±2.12 ab
*Clong95*	7.7 ± 0.35 bc	15.0 ± 0.52 b	26.7 ± 1.06 ab
*Clong500*	9.0 ± 0.71 ab	17.6 ± 0.31 a	28.6 ± 0.49 ab
*Amex50*	9.9 ± 0.85 ab	14.5 ± 0.10 b	24.7 ± 0.35 ab
*Amex95*	10.7 ± 0.35 a	17.5 ± 0.20 a	29.5 ± 2.12 a
*Amex500*	8.0 ± 0.71 bc	16.5 ± 0.10 ab	29.3 ± 0.92 a
*Cl50 + Am50*	8.3 ± 0.35 bc	15.0 ± 0.24 b	25.0 ± 1.41 ab
*p*-value	0.0018	0.0037	0.0136

Means with the same letter in the same column are not significantly different by Tukey’s test (*p* ≤ 0.05). Control refers to the absolute control group, *Clong50* stands for the CL_50_ of *C. longirostrata*, *Clong95* represents the CL_95_ of *C. longirostrata*, *Clong500* corresponds to a concentration of *C. longirostrata* of 500 mg/mL, *Amex50* stands for the CL_50_ of *A. mexicana*, *Amex95* represents the CL_95_ of *A. mexicana*, *Amex500* corresponds to a concentration of *A. mexicana* of 500 mg/mL, and *Cl50 + Am50* refers to a mixture of the CL_50_ of both extracts.

## Data Availability

The underlying data of this manuscript are available upon reasonable request from the authors.

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
