# Peer review of "Phytotoxicity of Extracts of Argemone mexicana and Crotalaria longirostrata on Tomato Seedling Physiology"

_plants, 2023, doi:10.3390/plants12223856_

Round 1

Reviewer 1 Report

Comments and Suggestions for Authors

The paper is interesting but it seems very preliminary. First of all, I disagree with the approach that "Phytotoxicity caused by secondary metabolites of botanical extracts is a drawback in agriculture." This is wrong if the phytotoxic effects are on the weeds and also does not fit well with the specific paper since the mixture promoted germination. English language revision is necessary (dat in capital letters, expressions like "produced greater increase" are totally wrong). Some more issues: a) materials and methods should be described in more details and include more measurements and not only germination, b) results should be discussed and authors are encouraged to readand write about the "hormesis" phenomenon and c) the discussion should be substantially enriched with the practical use of the results for agriculture. A major revision is strongly recommended.

Comments on the Quality of English Language

The help of an English native speaker is necessary

Author Response

1.- First, I disagree with the approach that "Phytotoxicity caused by secondary metabolites of botanical extracts is a drawback in agriculture."

Response:

It is important to understand that generalizing hormesis as a dose-response phenomenon causes the confusion that all chemical and organic compounds generate a negative effect. This can be observed in the review by Calabrese & Blain (2009, https://doi.org/10.1016/j.envpol.2008.07.028).

However, several studies have shown that a plant extract can generate an allelopathic effect on different species of agricultural interest, as demonstrated by the trial conducted by Miranda-Arámbula et al., 2021 (https://doi.org/10.17129/BOTSCI.2727). The objective of this study is to demonstrate the potential utilization of extracts of C. longirostrata and A. mexicana as plant growth promoters. Additionally, it is important to note that the doses utilized in our trial do not have a toxic effect on the same crop, thus avoiding the occurrence of hormesis in tomato plants.

Further information regarding this subject is included in the discussion. But, the study name was change to “Effect of Argemone mexicana and Crotalaria longirostrata extracts on tomato seedling physiology.

2.- Materials and methods should be described in more details and include more measurements and not only germination.

Response:

The sole consideration was the impact of the extracts on the germination of tomato seeds was considered, and no additional in vitro tests are currently available.

3.- Results should be discussed, and authors are encouraged to readand write about the "hormesis" phenomenon

Response: Comments are addressed and information is attached on the lines 268-279.

4.- The discussion should be substantially enriched with the practical use of the results for agriculture. A major revision is strongly recommended.

Response:

Comments are addressed and information is attached on the lines 280-299

Reviewer 2 Report

Comments and Suggestions for Authors

The present document describes the impact of two plants with potential phytotoxic effect on the germination and vigor of tomato seeds.

The introduction is clear and states the problem correctly, even if there is an ambiguity between the phytotoxicity and the potential beneficial effects the extracts can have.

The material and method are clearly presented, even I miss a control with only applying methanol to check the impact without any extract, because the only control made is with water.

The result section seems clear, but the text is really difficult to read, because there is no correlation between what it's written and what it's shown in the tables and figures, there is a big confusion between the real data and the relative that comparing situations that makes it difficult to understand.

The text swich from relative to absolute when the data (I suppose) are only relative because I cannot find these data in the tables/figures. This part should be rewritten enterely, and correcting in the same way the abstract which finally have the same problem, besides the difficulty to undestand some data presented.

Another problem that I'm not sure were you can solve without making the analysis (chemical characterisation of the compounds present in the two plants) is to stablish which is the variability of the composition and content on the molecules with have a potential activity on the germination and vigor of the seedlings. At least it should be discussed because your results depend on this characterisation to be complete.

More detailed comments can be found in the attached document

Author Response

1.- compared to what!

Response:

was modified as: plant height increased by 15.4% with the high dose of C. longirostrata (Clong500) compared to the control

2.- indicate what dat means at least once in the abstract

Response:

if indicated in line 21 as: days after transplanting (dat)

3.- not clear what you mean

Was modified as: Chlorophyll content increased with Amex95 by 66.1% in 10 dat, 22.6% at 20 dat, and 19.6% at 30 dat.

4.- in Itallic?

the reviewer's comment is noted

5.- please indicate in the legend what these names mean in order to be able to make and independent lecture of the table without reading the mat and method section

Response: The comment is noted and appended to all tables and figures: Control= refers to the absolute control group, Clong50= stands for the CL50 of C. longi-rostrata, Clong95= represents the CL95 of C. longirostrata, Clong500= concentration is 500 mg/mL of C. longirostrata, Amex50= stands for the CL50 of A. mexicana, Amex95= represents the CL95 of A. mexicana, Amex500= concentration is 500 mg/mL of A. mexi-cana, Cl50+Am50= refers to a mixture of the CL50 of both extracts.

6.- here what is written is that the treatment was made at 10 days, I suppose you want to say that 10 dat transplanting and being in contact witht he products.

Response:

Was modified as: 10 dat

7.- Besides which is the impact of the methanol alone on the germination capacity and vigour?

Response:

No methanol treatment was implemented in the experiment.

8.- I do not read this in table 4

Response: 

is modified as: (10.7 mm, 17.5 mm, and 29.5 mm, respectively), line 137

9.- I do not read this in table 4

Response:

Was modified as: (10.7 mm, 17.5 mm, and 29.5 mm, respectively), line 137

10.- same thing

Response:

Was modified as: (17.6 mm), line 139

11.- same

Response:  

is removed line 139

12.- same

Response:  

Was modified as: 29.5 mm and 29.3 mm, line 140

13.- data do not correspond to the figure

Response: 

is modified as: (44.9 SPAD and 45.1 SPAD, respectively)

14.- with respect to?

Response:

Was modified as: compared to the control, lines 152-153.

15.- lower to what?

Response:

is modified as: but the value was higher in all treatments compared to the first evaluation, lines 153-154.

16.- I cannot see this data in the figure where values are below 20

Response:

Was modified as: Clong500 with 58.5 SPAD and Amex95 with 58.7 SPAD, line 154.

17.- difficult to say. I do not know if you are talking about the data, the differences between the two measurements, really confusing as no data have a coincidence with the figure, and previously with the tables

Response:

Was  modified as: The percentage was replaced by SPAD and nitrogen values, as mentioned in lines 155-158.

18.- no, it does not reached this, maybe in relative value but the sentece do not express that

Response: 

Was modified as: The percentage was replaced by grams, lines 170-178.

19.- can you explain the acronym?

Response:

Was modified as: days after emergency (dae)

20.- It will be interesting to know the variability of the compounds contained by the two plants that may have phytotoxic effects to judge about the pertinence of your study, to be placed in the introduction.

Response: 

Added information in lines 77-86.

21.- Another problem that I'm not sure were you can solve without making the analysis (chemical characterisation of the compounds present in the two plants) is to stablish which is the variability of the composition and content on the molecules with have a potential activity on the germination and vigor of the seedlings. At least it should be discussed because your results depend on this characterisation to be complete.

Resposne:

It is mentioned in the lines 84-86, 250-255. The analysis of the extracts was previously published in https://doi.org/10.17268/sci.agropecu.2022.007 y https://doi.org/10.15741/revbio.10.e140

Round 2

Reviewer 1 Report

Comments and Suggestions for Authors

Authors have addressed themajority of the raised issues and therefore I recommend the acceptance of their manuscript

Comments on the Quality of English Language

Adequate

Reviewer 2 Report

Comments and Suggestions for Authors

The authors have now provided all the requested changed and now the results are comprehensible and clear.